

# Source apportionment study on particulate air pollution in two high-altitude Bolivian cities: La Paz and El Alto

Valeria Mardoñez[1,2], Marco Pandolfi[3], Lucille Joanna S. Borlaza[1], Jean-Luc Jaffrezo[1], Andrés Alastuey[3], Jean-Luc Besombes[4], Isabel Moreno R.[2], Noemi Perez[3], Griša Močnik[5,6,7], Patrick Ginot[1], Radovan Krejci[8], Vladislav Chrastny[9], Alfred Wiedensohler[10], Paolo Laj[1,11], Marcos Andrade[2,12], Gaëlle Uzu[1]

[1] Institute des Géosciences de l'Environnement, Université Grenoble Alpes, CNRS, IRD, Grenoble INP, Grenoble, France.
[2] Laboratorio de Física de la Atmósfera, Instituto de Investigaciones Físicas, Universidad Mayor de San Andrés, La Paz, Bolivia.
[3] Institute of Environmental Assessment and Water Research (IDAEA-CSIC), Barcelona, 08034, Spain
[4] Université Savoie Mont Blanc, CNRS, EDYTEM (UMR 5204), Chambéry 73000 France
[5] Center for Atmospheric Research, University of Nova Gorica, 5270 Ajdovščina, Slovenia
[6] Haze Instruments d.o.o., 1000 Ljubljana, Slovenia
[7] Department of Condensed Matter Physics, Jozef Stefan Institute, 1000 Ljubljana, Slovenia
[8] Department of Environmental Science & Bolin Centre for Climate Research, Stockholm University, 10691 Stockholm, Sweden
[9] Department of Environmental Geosciences, Faculty of Environmental Sciences, Czech University of Life Sciences Prague, Kamýcká 129, 165 00, Prague-Suchdol, Czech Republic
[10] Leibniz Institute for Tropospheric Research (TROPOS), 04318 Leipzig, Germany
[11] Institute for Atmospheric and Earth System Research (INAR), University of Helsinki, 00014 Helsinki, Finland
[12] Department of Atmospheric and Oceanic Sciences, University of Maryland, College Park, MD, USA

*Correspondence to*: Valeria Mardoñez (valeria.mardonez@univ-grenoble-alpes.fr)

**Abstract.** La Paz and El Alto are two fast-growing high-altitude Bolivian cities forming the second largest metropolitan area in the country, located between 3200 and 4050 m a.s.l. Together they host a growing population of around 1.8 million people. The air quality in this conurbation is strongly influenced by urbanization. However, there are no comprehensive studies that have assessed the sources of air pollution and their impacts on health. Despite being neighboring cities, the drastic change in altitude and topography between La Paz and El Alto together with different socio-economic activities lead to different sources, dynamics and transport of particulate matter (PM). In this investigation, $PM_{10}$ samples were collected at two urban background stations located in La Paz and El Alto between April 2016 and June 2017. The samples were later analyzed for a wide range of chemical species including numerous source tracers (OC, EC, water-soluble ions, sugar anhydrides, sugar alcohols, trace metals, and molecular organic species). The US-EPA Positive Matrix Factorization (PMF v.5.0) receptor model was then applied for source apportionment of $PM_{10}$. This is the first source apportionment study in South America that incorporates a large set of organic markers (such as levoglucosan, polycyclic aromatic hydrocarbons - PAH, hopanes and alkanes) together with inorganic species. The multisite PMF resolved 11 main sources of PM. The largest annual contribution to $PM_{10}$ came from two major sources: the ensemble of the four vehicular emissions sources (exhaust and non-exhaust), together responsible for 35% and 25% of the measured PM in La Paz and El Alto, respectively, and dust contributing 20% and 32% to the total. Secondary aerosols contributed 22% (24%) in La Paz (El Alto). Agriculture-related smoke from biomass burning originated

in the Bolivian lowlands and neighboring countries contributed to 8% (7%) of the total $PM_{10}$ mass annually. This contribution increased to 17% (13%) between August-October. Primary biogenic emissions were responsible for 13% (7%) of the measured $PM_{10}$ mass. Finally, it was possible to identify a profile related to open waste burning occurring between the months of May

and August. Despite the fact that this source contributed only to 2% (5%) of the total $PM_{10}$ mass, it constitutes the second largest source of PAHs, compounds potentially hazardous to health. Our analysis resulted in the identification of two specific traffic related sources. In addition, we also identified a lubricant source (not frequently identified) and a non-exhaust emissions source. This study shows that $PM_{10}$ concentrations in La Paz and El Alto region are mostly impacted by a limited number of local sources. In conclusion, dust, traffic emissions, open waste burning and biomass burning are the main sources to target in

order to improve air quality in both cities.

# 1 Introduction

Outdoor air pollution has undeniably proven to be an important threat for public health, being responsible for about 4.2 million yearly premature deaths around the world every year (WHO, 2021). The exposure to air pollution becomes more complex at higher altitudes due to the decrease in oxygen per volume of air since people have developed a higher lung capacity in order

to fulfill the body oxygen demand (Frisancho, 1977; Frisancho, 2013; Frisancho et al., 1999; Madueño et al., 2020; U.S. EPA, 2011).

Many of the high-altitude large cities in the world (> 2000 m a.s.l., ≳2 million inhabitants) are located in Latin American low and middle-income countries, among which are Mexico City, Bogotá and Quito. These cities are also subject of a developing industry and a growing vehicular fleet that translates in constantly increasing energy consumption that is grounded on non-

renewable energy sources (Castro Verdezoto et al., 2019; Molina et al., 2019; Pardo Martínez, 2015). Most of the cities in the region with available data face a deteriorated air quality, with particulate matter (PM) concentrations that exceed the World Health Organization (WHO, 2021) guidelines (Gutiérrez-Castillo et al., 2005; Mugica et al., 2009; Ramírez, et al., 2018; Zalakeviciute, et al., 2020).

High-altitude cities share some specific characteristics due to complex topography and related meteorology, influencing

transport, accumulation and dispersion of air pollution. In addition, high altitude is linked to strong solar radiation and thus favoring photochemical activity and high daily temperature variations. Compared with other regions at similar latitudes, high altitude cities in South America experience lower temperature, lower atmospheric pressure and thus related lower saturation vapor pressures, more complex wind patterns and reduced precipitation (Vega et al., 2010; Zalakeviciute et al., 2018). Previous studies have shown that these specific high-altitude atmospheric and thermodynamic conditions can strongly favor new aerosol

particle formation (NPF) (Boulon et al., 2010; Brines et al., 2015; Hallar et al., 2011; Sellegri et al., 2019; Singla et al., 2018; Sorribas et al., 2015). It has also been observed that low oxygen environments alter the performance and reduce the efficiency of combustion engines (Martínez et al., 2022; X. Wang, et al., 2013a). This, consequently, changes the vehicular emissions of



gaseous and particulate pollutants (Bishop et al., 2001; Giraldo & Huertas, 2019; He et al., 2011; Nagpure et al., 2011; X. Wang et al., 2013b).

Listed amongst the highest metropolitan areas in the world, La Paz (between 3200-3600 m a.s.l.) and El Alto (4050 m a.s.l.) are two Bolivian cities that form part of the same conurbation, with a population of approximately 1.8 million people. Despite their proximity, there are large topographical, meteorological and socio-economic differences between them. Although concentrations of some pollutants are regulated by the Bolivian law (CO, $SO_2$, $NO_2$, $O_3$, TSP, $PM_{10}$, Pb[1]), very few air quality studies which include long term measurements at moderate time resolution have been performed in the country or in the region.

Few previous studies reported $PM_{10}$ mass concentrations between 10 and 100 µg m$^{-3}$ measured in urban and urban-background stations in La Paz and El Alto (Red MoniCA, 2016, 2017, 2018; Wiedensohler et al., 2018). However, no particle chemical speciation was ever performed that would permit identifying the major sources responsible for the high PM concentrations. Moreover, measurements performed in the nearby Chacaltaya GAW station (CHC-GAW: 16.350500°S, 68.131389°W, 5240 m a.s.l.) show that the emissions of the city not only have a local impact, but can also act as a point source influencing

atmospheric composition at regional scale (Aliaga et al., 2021).

While little is known about the sources of PM in the country, since industry is not largely developed, vehicular emissions potentially represent an important source of air pollution, particularly considering the lack of restrictions regarding the age of the vehicle fleet. It has been reported that 43% of the circulating vehicles are less than 10 years old, 15% are 10-20 years old and another 24% are 20-30 years old (INE, 2020a, 2020b). At a regional scale, agricultural biomass burning in the Bolivian

and Brazilian valleys and rain-forests constitutes an important seasonal source of particulate pollutants (Mataveli et al., 2021). The latter has a significant impact on the air quality of the cities close to where the fires take place (Nawaz & Henze, 2020) and can be transported over large distances. Studies have shown that air masses coming from the Amazon can travel and carry pollutants across the Andes, thus reaching the Bolivian Altiplano (Bourgeois et al., 2015; Chauvigne et al., 2019; Magalhães et al., 2019; Segura et al., 2020). In addition, previous studies based on emission inventories adapted to the data availability in

LP-EA pointed out road dust, food industry, cooking and vehicle emissions as the major sources of $PM_{10}$, whereas for Cochabamba (the third largest urban area in Bolivia) estimations showed mobile sources to be responsible for almost 90% of $PM_{10}$ emissions (Herbst, 2007; Pareja et al., 2011). Although there are some indications of the most outstanding sources of particulate matter in La Paz and El Alto, currently there is not comprehensive study on the composition and sources of particulate matter air pollution. Hence, the aim of this study is to apportion and characterize the sources of PM that affect air

quality in the metropolis of La Paz-El Alto, which can be used as a baseline for future policy making.

To achieve the goal described above, we applied the EPA-Positive Matrix Factorization (PMF v.5.0) receptor model on the $PM_{10}$ chemical speciation obtained from 24-h filter samples collected simultaneously in La Paz and El Alto during a 15-months campaign. This is one of the very few studies characterizing PM in Bolivia over an extended time period. With limited studies in this region, identifying the sources and chemical profiles of PM in the study sites proved to be more challenging. A

---

[1] A more extensive description of the permissible air quality limits policies can be found in the supplementary material (SI)



comprehensive chemical speciation was included in the analysis: ionic species, monosaccharide anhydrides, polyols, metals, PAHs, alkanes, and hopanes. To the best of our knowledge, this is the first study on source apportionment at high-altitude cities which include such a large set of organic and inorganic species.

## 2 Method

### 2.1 Sampling sites

The two study sites, La Paz (LP) and El Alto (EA), possess significant topographical differences, apart from the important difference in altitude and pressure. While the city of El Alto lies on the Altiplano plateau, an open and flat area, the city of La Paz sprawls along the mountain valleys formed below the Altiplano in a closed area with steep and complex topography. Although the characteristic tropical seasonal change between a dry and a wet season governs the meteorological conditions throughout the year, temperature and wind patterns differ between both cities largely due to the local topography.

Furthermore, the city of El Alto began as a peri-urban zone of the city of La Paz, welcoming migrants from nearby towns and communities who moved to the outskirts of the city of La Paz (Fernández, 2021). This gave rise to large economic and social differences between the cities that to some extent still persist and are observable among the general population (Foster & Irusta, 2003) . Such differences could have an impact on air pollutant emissions associated to the different practices in each of the cities together with the daily commute of a significant part of the population of El Alto towards the city of La Paz.

The few existing industries are mostly found within or in the surroundings of El Alto, and the vehicular fleet observed in both cities is not homogeneous. The density of heavy vehicle traffic – trucks and buses, is higher in El Alto, since it is the main regional and international connection from and to the metropolis. All these factors uphold the need for having independent representative sampling sites for each city rather than just one, in spite of them being part of the same conurbation.

The sampling campaign was carried between April 2016 and June 2017. Several ambient and meteorological parameters were

measured simultaneously at two urban background sites, one in each city. The sampling sites were located 7 km apart, with an altitude difference of more than 400 m, and at approximately 20 km distance from the Chacaltaya Global Atmosphere Watch (CHC-GAW) monitoring station (Figure).

The El Alto measurement site was installed within the El Alto International Airport, in the facilities of the meteorological observatory (16.5100° S, 68.1987° W, 4025 m a.s.l.). The observatory was at a distance of approximately 250 m from the

airport runway and 500 m from the nearest major road and has been described elsewhere (Wiedensohler et al., 2018). Pre-campaign measurements were made to assess if the airplanes takeoff and landing influenced the measurements, finding no significant influence in $CO_2$, $PM_1$ and $PM_{2.5}$ at each airplane arrival and departure. Road traffic within the airport perimeter is almost non-existent. The area around the sampling site is unpaved, hence dusty, and there are no other buildings in the proximity of the observatory. In March 2016, just prior the beginning of the sampling, the airport administration cleared the

ground within the perimeter fence of the meteorological observatory, leaving the site dustier than the rest of the airport.





La Paz measurement site (LP) was placed on the rooftop of the city's Museum Pipiripi (Espacio Interactivo Memoria y Futuro Pipiripi: 16.5013°S, 68.1259°W, 3600 m a.s.l.). This municipal building is located on a small hilltop downtown La Paz. Unlike the EA site, within a 1 km radius, the LP site is surrounded by many busy roads and dense residential areas, with a horizontal and vertical minimum distance to the nearest road of approximately 70 and 45 m respectively. Otherwise, the site's immediate

surroundings (~100 m radius) are covered by green areas and a municipality buses parking lot at the base of the hill.

## 2.2 Sampling methods

High-volume samplers (MCV CAV-A/mb with an MCV PM1025UNE (PM10) head) were used to collect 24-h filter samples of PM every third day at both sites. Sampling started at 9:00 a.m. and the flow was automatically kept at 30 $m^3$ $h^{-1}$. The samplers were placed on the rooftop of the buildings in order to avoid interference of near-ground particle resuspension. During

the period analyzed in the present study, an impactor with a 50% collection efficiency of aerosol particles with an aerodynamic equivalent diameter of 10 µm was placed at the inlet of the samplers to provide an upper size-cut at both sites.

The mass concentrations measured at both sampling sites are hereafter reported in ambient conditions (EA: $\overline{T} = 280.8\,K$, $\overline{P} = 628.2\,hPa$, LP: $\overline{T} = 286.0\,K$, $\overline{P} = 664.7\,hPa$), unless stated otherwise (e.g. when compared to literature reported concentrations). In order to convert to standard conditions of temperature and pressure ($\overline{T} = 273\,K$, $\overline{P} = 1013.5\,hPa$) the

concentrations must be multiplied by a factor of 1.66 and 1.60 in El Alto and La Paz, respectively. Since the difference in ambient concentrations between the sites due to a difference in mean temperature and pressure is of approximately 4%, ambient concentrations are directly compared between the sites in the following sections.

The aerosol particles were collected onto pre-heated (8 hours at 500°C) and pre-weighted 150 mm-diameter quartz fiber filters (Pallflex 2500QAT-UP). After sampling, the filters were folded and wrapped in aluminum foil, sealed in impermeable plastic

bags, and stored in a cool environment before being transported for analysis. Mass concentrations were first measured gravimetrically and then the samples were divided for chemical analysis in three European laboratories. The resulting chemical speciation comprised of elemental carbon (EC), organic carbon (OC), sugar anhydrides (Levoglucosan, mannosan), sugar alcohols (arabitol, mannitol), water soluble ions ($SO_4^{2-}$, $NO_3^-$, $Cl^-$, $MSA^-$, $NH_4^+$, $Na^+$, $K^+$, $Mg^{2+}$, $Ca^{2+}$) measured at IGE, Grenoble, France; metals (Al, Ca, K, Na, Mg, Fe, Ti, V, Mn, Cu, Zn, Rb, Sn, Sb, Pb) measured at IDAEA, CSIC, Barcelona,

Spain; Polycyclic aromatic hydrocarbons (PAHs: Fla, Pyr, Tri, BaA, Chr, BaP, BghiP, IP, BbF, Cor), alkanes (C21-C26), methyl PAHs, thiophens, hopanes (HP3-HP4) alkane methoxyphenols, and methylnitricatechols measured at EDYTEM, Chambéry, France[2]. A total of 92 and 103 filter-samples were collected in the cities of El Alto and La Paz, respectively, excluding samples having sampling flow issues or influenced by specific events (c.a. San Juan local festivity, Christmas, New Year's Eve). In addition, laboratory blank filters were used to calculate the limits of quantification (QL). The average

concentrations measured from the laboratory-blanks were then subtracted from the samples measured atmospheric concentrations.

---

[2] A more exhaustive and detailed table of all the measured species and the analysis methods can be found in the SI.





## 2.3 Source apportionment (PMF)

The Positive Matrix Factor PMF 5.0 tool (Norris & Duvall, 2014; Paatero & Tapper, 1994), developed by the U.S. Environmental Protection Agency (EPA), was used to apportion the sources that contribute to the observed particulate material in the collected samples at both sites. This non-negative multivariate factor analysis seeks to solve the chemical mass balance equation [1], applying a weighted least-squares fit algorithm, $x_{ij}$ representing each of the elements of the concentration matrix (having $n$ number of samples and $m$ number of chemical species measured), $g_{ik}$ are the contributions of each $k$ factor to the $i$th sample, $f_{kj}$ are the chemical profile of the factors, and $e_{ij}$ are the residuals (i.e. the difference between the calculated and the measured concentration).

$$x_{ij} = \sum_{k=1}^{p} g_{ik} f_{kj} + e_{ij} \qquad (1)$$

The optimal solution is then achieved by minimizing the function Q defined as:

$$Q = \sum_{i=1}^{n} \sum_{j=1}^{m} \left[ \frac{x_{ij} - \sum_{k=1}^{p} g_{ik} f_{kj}}{u_{ij}} \right]^2 \qquad (2)$$

where $u_{ij}$ are the uncertainties associated to each measurement.

### 2.3.1 Sample and chemical species selection

Out of the 197 $PM_{10}$ samples initially included, 12 of them were later excluded from the analysis for having over 6 species with missing values (EA: 19 Sep 2016, 11 Jan 2017; LP: 14 May 2016, 07 Jun 2016, 12 Dec 2016, 02 May 2017) or because they presented unusual concentrations of PM or several species (LP: 04 Apr 2016, 22 May 2017, 30 May 2017, 11 Jun 2017, 15 May 2017, 19 Jun 2017). For each filter, 178 chemical species were measured. The species that presented irregularities in their time series were excluded from the analysis together with the ones that had over 25% of the data below the quantification limit (<QL, defined as the mean field-blank concentrations measured per specie, plus two times the standard deviation). From the remaining 86 species, the ones that were measured through both Ion Chromatography (IC) and Inductive Coupled Plasma-Mass Spectrometry (ICP-MS), only the ICP-MS metals were included in order to avoid double counting, except for $K^+$, for which the IC measurements were used since water soluble $K^+$ is a known tracer for biomass Burning (BB) (Li et al., 2021). Galactosan and sorbitol were considered unnecessary tracers for biomass burning and primary biogenic aerosols, respectively, given the presence of other specific tracers as levoglucosan, mannosan, mannitol and arabitol. Thus, they were excluded from the analysis. Finally, other non-specific-tracer metal species were excluded after several attempts of including them in the PMF input data, since they proved to only add instability to the solution. Based on the results of Samaké (2019a), arabitol and mannitol were added as one Polyol-representative specie, given that they are emitted by the same source and have a Pearson correlation of r>0.7, for both sites. The same was done for PAHs that presented a r>0.9 (PAH_1: [BghiP]+[IP]+[BbF]; PAH_2: [Fla]+[Pyr]). Lastly, OC was replaced in the PMF analysis by OC*, which is defined as the subtraction of the carbon mass concentration of all the included organic compounds out of the measured OC concentrations, to avoid double counting (e.g. Weber et al., 2019):





$$OC^* = OC - \begin{pmatrix} 0.12 \cdot [MSA] + 0.40 \cdot [Polyols] + 0.44 \cdot ([Levoglucosan] + [Mannosan]) + \\ 0.95 \cdot ([BghiP] + [IP] + [BbF] + [Fla] + [Pyr] + [BaA] + [Chr] + [Tri] + [BaP] + [Cor]) + \\ 0.85 \cdot ([C21] + [C22] + [C23] + [C24] + [C25] + [C26]) + \\ 0.87 \cdot ([HP3] + [HP4]) \end{pmatrix} \quad (3)$$

**2.3.2 Uncertainty calculation and specie weight-assignment**

For the uncertainty matrix, a 10% uncertainty was assigned to PM mass measurements, used as the total variable in the PMF. The uncertainty calculation for polyols, monosaccharide anhydrides, and ions was performed using the formula proposed by Gianini (2012), using the variation coefficients (CV) and the additional coefficients of variation (a) proposed and used by Weber (2019), with the average QL associated to each specie instead of DL. The uncertainties associated to EC, OC, and metals, were calculated following the method proposed by Amato (2009) and Escrig (2009). Finally, the uncertainties assigned to the molecular organic species were calculated using the formulas proposed by Polissar (1998) and Reff (2007), replacing the DL values by QL.

Values under the QL in the concentration matrix were replaced by the average of QL divided by 2, for each specie, and the corresponding uncertainties were set to $\frac{5}{6}$ QL (Norris & Duvall, 2014). The outliers encountered in the time series of some species (a total of 4 values) were replaced by NA. Then, the missing values in the input file were set to be replaced in the software by the median value of the corresponding specie and their associated uncertainty was automatically set to four times the species-specific median.

The weight of the species in the factor analysis was determined based on their signal to noise ratio (S/N). Species with a S/N>2 were defined as strong. Species with a signal to noise ratio: 0.2≤ S/N≤2 were defined as weak, which down weighs its influence in the analysis by triplicating their uncertainties. Species with a S/N<0.2 were not included in the analysis. Finally, PM was set as total variable, automatically setting it as a weak variable. After several tests, some variables were also set as weak (K⁺, V), because setting them as strong would create a separate specific factor without any relevant meaning. The PAHs, alkanes and hopaes were set as weak species, to prevent them from driving the solution.

**2.3.3 Solution evaluation criteria**

Solutions ranging from 8 to 13 factors were explored, in order to select the appropriate number of factors contributing to each site. A series of statistical and geochemical control parameters were then evaluated in order to choose the final solution (Belis C.A. et al. 2019):

- $Q_{true}/Q_{robust} < 1.5$.
- Residuals per specie centered and symmetrically distributed around 0, and within -3 and 3 (with the exception of a few outliers).
- Bootstrap (BS) evaluation of the statistical robustness of the selected base run having a correlation coefficient for every factor > 0.8 after 100 iterations, before and after constraints.



- Displacements (DISP) analysis evaluating the rotational ambiguity and tolerance of the solution to small perturbations (No observed rotation was observed for dQmax = 4, 8).
- Geochemical consistency of the obtained factor chemical profiles based on literature and knowledge of the study site.

### 2.3.4 Multisite PMF

Single-site PMF analysis were initially run in parallel, showing indeed similar main sources contributing to particulate matter. Increasing the number of factors showed a promising possibility of splitting the traffic profile but at the cost of altering the statistical stability of the solution. These results were a motivation to run a multisite PMF. Such approach has proven to reduce the rotational ambiguity in factor analyses (Dai et al., 2020; Hernández-Pellón & Fernández-Olmo, 2019; Hopke, 2021; Pandolfi et al., 2020), increasing the statistical robustness while increasing the number of samples. For this purpose, in order to combine both datasets as one (EA-LP) the dates of the La Paz dataset were shifted in time by two years and then appended to El Alto's dataset, thus avoiding repeated dates and composing a single input matrix for PMF that respected the natural seasonal variability of the original datasets. The multisite approach stands on the hypothesis that the major sources contributing to $PM_{10}$ in both sites are similar and display similar chemical profiles, which has been verified within the single site solutions.

### 2.3.5 Set of constraints

Once the optimum number of factors was selected in the multisite base solution, a set of "soft" constraints (Table 1) was applied to the selected solution based on previous studies (Borlaza et al., 2021; Samaké, et al., 2019b; Weber et al., 2019) Table1

### 2.3.6 Additional analysis of one local specific source: fuel chemical fingerprint

In order to further investigate the differences between the two main types of fuel used in LP-EA, 3 samples of both gasoline and diesel were taken at 3 randomly chosen gas-stations located in different parts of the city. The samples were analyzed for main metal composition as follows: 1 ml of sample (gasoline, diesel) was transferred into a Teflon microwave vessel (Anton Paar microwave laboratory unit). Then, 10 ml of $HNO_3$ (double distilled, suprapure level) were added and the solution was decomposed by increasing temperature and pressure (175°C and 10 bar). In the microwave, the EPA 3051A method was run twice to assure that the solutions were indeed decomposed (USEPA, 2007). After cooling down the vessels, the solutions were diluted by a factor of 10 and directly measured using inductively coupled plasma mass spectrometer (ICP-MS). A complete descriptive table of the analyzed species can be found in the SI.



## 3 Results

### 3.1 Seasonal variations of chemical components of PM$_{10}$

A yearly alternation between the dry and the wet season as presented in Figure, shows an annual maximum of PM$_{10}$ concentrations coinciding with the middle of the dry season (Southern hemisphere winter). During this season, almost no wet deposition takes place and favorable conditions for particle resuspension are common. Maximum ambient PM$_{10}$ daily concentrations of $37.2 \pm 10.5$ µg m$^{-3}$ and $33.2 \pm 7.5$ µg m$^{-3}$ were measured during this period (May-August) in El Alto and La Paz, respectively. The opposite is observed during the wet season (Southern hemisphere summer, December-March), where

precipitation events are very frequent and daily minimum-temperatures reach their highest values.

Similar variability and concentrations of PM$_{10}$ were measured at the International Airport of El Alto, using the C$^{14}$ beta-attenuation technique, between 2011 and 2015 (ranging between ca. 10-50 µg m$^{-3}$ throughout the year, Red MoniCA, 2016a). In the case of La Paz, the variability reported by city's Municipal Secretary of Environmental Management (MSEM) using the C$^{14}$ beta-attenuation technique was similar to the one observed in the present study. However, the reported PM$_{10}$ concentrations

were higher (Red MoniCA, 2016, 2017, 2018). The difference in the measured concentrations in the case of La Paz can probably be explained by the different locations of the measurement sites since the site of MSEM measurements is located in the downtown area, next to a busy avenue.

From all collected samples during the campaign, 5 and 12% of the daily samples collected in La Paz and El Alto exceeded, respectively, the 24-hour PM10 concentration of 45 µg m-3, not to be exceeded more than 3-4 days per year according to the

short-term PM10 Air Quality Guideline (AQG) level recommended by the World Health Organization (WHO, 2021). Moreover, the annual PM10 concentrations in both cities are at least 1.2 times higher than the PM10 levels of 15 µg m$^{-3}$ recommended as annual AQG by the same organization (WHO, 2021). Average measured PM$_{10}$ concentrations were found to be $29.9 \pm 12.0$ µg m$^{-3}$ (STP: $49.6 \pm 19.9$ µg m$^{-3}$) in El Alto and $27.2 \pm 8.9$ µg m$^{-3}$ in La Paz[3] (STP: $43.5 \pm 14.2$ µg m$^{-3}$). However, the annual average values can be relatively lower due to the under sampling during the wet season.

The observed concentrations are lower compared to what was reported for Mexico City, a high-altitude (2850 m a.s.l) Latin-American megacity (Table 2), higher to what was observed in Bogotá, and comparable, in the case of La Paz, to what was reported for Quito. Nevertheless, Quito is the only one comparable in terms of population density. Moreover, average concentrations found in La Paz-El Alto are almost twice the reported average concentrations for suburban and urban background sites in Europe with average normalized PM$_{10}$ concentrations comparable to what was measured in Turkey, some

regions in Poland (Rybnik: 44.1 µg m$^{-3}$), Bulgaria (Vidin: 41.3 µg m$^{-3}$), North Macedonia (Skopje: 48.7 µg m$^{-3}$) and Italy (Napoli: 46.9 µg m$^{-3}$) in 2019 (EEA, 2020; EEA, 2022).

The reconstruction of the measured PM$_{10}$ mass resulted from the mass closure procedure described for organic matter in Favez (2010) and Putaud (2004); non-sea-salt sulfate in Seinfeld & Pandis (1998) and dust in Alastuey (2016) and Pérez (2008):

---

[3] The statistics presented hereafter refer only to the period in which measurements were made, and to the samples collected during that period.



$$PM(recons) = (1.8 \cdot [OC]) + [EC] + ([SO_4^{2-}] - 0.252 \cdot [Na^+]) + [NO_3^-] + (1.89 \cdot [Al]) + (3 \cdot (1.89 \cdot [Al])) + (1.5 \cdot$$

$[Ca]) + [Ca] + [Fe] + [K] + [Mg] + [Mn] + [Ti] + [P]$                    (4)

Average $PM_{10}$ (recons.) [4]/ $PM_{10}$ (meas.) ratios of 0.86 in El Alto and 0.77 in La Paz. The remaining unidentified mass fraction may be attributed to the loss of volatile organic matter and secondary aerosols after the post-weighing, throughout the transport until the analysis of the samples. A 10% uncertainty associated with the PM mass measurements could also have a role in the observed difference.

The total average partition of the chemical species that significantly contribute to the measured $PM_{10}$ concentrations in El Alto during the campaign was: 22±5% OM (i.e. $1.8 \cdot OC$), 5±2% EC, 9±5% the sum of secondary inorganic aerosols ($NH_4^+, NO_3^-$, and $SO_4^{2-}$), and 12±3% of crustal material (Al, Fe, Ti, Ca, K, Mg, Mn, P). In La Paz, 25±5% OC, 6±2% EC, 8±5% the sum the secondary inorganic aerosols, and 10±2% of crustal material. Moreover, figS3 in the SI shows the monthly behavior of the principal species contributing to PM, together with some specific source tracers.

Mean OC/EC mass ratios of 2.6±1.1 and 2.8±1.6 were found for El Alto and La Paz, respectively, during the measurements period. This average OC/EC ratio results from the combination of different sources such as vehicle emissions together with other primary and secondary local and regional sources of carbonaceous particles (such as biomass burning, primary biogenic emissions and secondary organic aerosols). Highest OC/EC ratios with the largest standard deviation were obtained between August and October, peaking in September. The mean OC/EC ratios during this period is of 3.5±1.3 for El Alto and 3.8±1.6,

pointing out not only to the long-range influence of biomass burning emissions at the end of the agricultural year, but also to the influence of primary organic emissions (Mariola Brines et al. 2019; Hays et al. 2002; Robert et al. 2007; Robert, Kleeman, and Jakober 2007; Samaké, et al. 2019a,b; Waked et al. 2014). It was observed that biomass burning tracers peak in August, while polyols display an increase in concentrations peaking in September. In contrast, minimum OC/EC ratios that display a smaller dispersion around the mean were observed between March and April: 1.9±0.6 and 2.0±0.6 in El Alto and in La Paz,

respectively.

### 3.2 Source apportionment

After approaching the analysis individually for each site and seeing that both sites shared similar sources and considering the proximity of both cities, the multisite approach allowed us to overcome the difficulty of the fair low number of samples compared to the number of species included in the analysis. This immediately provided a solution with greater stability,

maintaining the previously observed profiles and making it possible to achieve a stable 11-factor solution. Figure3 displays the percentage contribution attributed by the PMF analysis to each of the resolved sources after applying the constraints described in the previous section.

---

[4] A sea salt term was excluded from the equation due to the little influence from marine aerosols sources in this Mediterranean site. Moreover, an important source of chloride related to anthropogenic activities was observed, misleading the reconstruction of sea salt.



The modeled $PM_{10}$ versus the measured $PM_{10}$ concentrations through the multisite approach presented a linear behavior with a slope of 1.01 and an $R^2$=0.95, meaning the factor analysis was capable to reproduce adequately the measured concentrations.

The 11 resolved sources include: dust, secondary sulfate, secondary nitrate, primary biogenic aerosols (PBA), MSA-rich, biomass burning (BB), traffic 1 (TR2), traffic 2 (TR2), lubricant, non-exhaust emissions, and waste burning (Fig. 3). Most of the resolved sources are consistent with the emission sources observed in previous studies performed in other sites (Chevrier, 2016; Waked et al., 2014; Weber et al., 2019; H. Yang et al., 2016). A comparison of the chemical profile of the sources resolved in the present study, and the chemical profile of the sources resolved by Borlaza (2021) and Weber (2019) using the

PD-SID method described in Belis (2015) and Pernigotti & Belis (2018) can be found in the SI. In addition, a separation of the traffic exhaust emissions (TR1, TR2) linked to the type of fuel used will also be presented in the following sections.

Dust and the ensemble of vehicular contributions (i.e. Traffic 1, Traffic 2, Lubricant, Non-exhaust emissions) are together responsible for 55% and 57% of the measured $PM_{10}$ mass concentrations in La Paz and El Alto. The dust factor has outstanding contributions of 32% in the city of El Alto, becoming the dominant source in this city. For La Paz, the vehicular emissions

take the lead in terms of percentage contributions (35%). The factors associated to secondary aerosols factors were responsible for nearly 22% and 24% of total PM (La Paz and El Alto respectively), only a slight difference can be observed between the cities except for the nitrate rich profile. Finally, the biomass burning factor was responsible for an average of 9 and 8% of the total measured $PM_{10}$ (in LP and EA, respectively) The chemical profiles and seasonality of each factor are displayed in Fig. 4 and Fig. 5, respectively, and will be discussed in more detail later.

In our case, among the advantages of performing a multisite PMF is the possibility of differentiating between two traffic profiles that could hardly be observed in the individual solutions. Similarly, some factor profiles that remained mixed in the single-site-solution for one site were polished as a result of the combination of both datasets. That was the case for the dust, MSA-rich, traffic 2, and non-exhaust profiles (Single site solutions can be found in the SI for comparison with the multisite solution.

**3.2.1 Dust**

This factor is the major contributor to the observed $PM_{10}$ mass at both sites and is traced by crustal elements, such as Al, Fe, Ti, Mg, Mn, Ca, Na, K, V, Rb. The confidence interval for these species is very small around the average displacement value, which means that these species are mainly the ones that define this source profile. The presence of other elements such as sulfate, OC, Zn and Pb (with tight confidence interval), together with EC, Cu (with confidence intervals that allow negligible

concentrations), supports the influence of road traffic in this source, through road dust resuspension. This factor has an average contribution of 32% (10.6±7.6 µg m⁻³) to the total $PM_{10}$ mass observed in El Alto during the measurements period, and 20% (5.5±4.1 µg m⁻³)[5] in the city of La Paz. This factor is largely responsible for the difference in PM mass concentrations observed

---

[5] Normalized concentrations to standard conditions of temperature and pressure: EA: 15.7±11.2 µg m⁻³; LP: 8.0±5.7 µg m⁻³





between La Paz and El Alto. This factor contribution can rise up to 46% of the mass in El Alto during winter time (specifically in June), whereas its percentage contribution in La Paz reached their maximum during the transition month of October (27%).

The difference in contribution between these two sites can be attributed to difference in La Paz and El Alto characteristics. Particularly, El Alto is a fast-growing city located on the edge of the Altiplano region, a dry and arid area with mostly unpaved streets and active construction works. On the other hand, the city of La Paz shows to be less influenced by this factor, likely because it has a higher fraction of paved roads compared to El Alto. In addition, La Paz is situated at a lower elevation, surrounded by mountains and hillsides, which reduces the impact of strong winds from the Altiplano. Although both stations

were considered to represent urban background, the terrain surrounding the two stations is very different. The El Alto station is located in the middle of the airport facilities, in a rather dusty area, while the La Paz station is located on the rooftop of a building located in the middle of the city. Nevertheless, combining the time series obtained from the PMF for this factor and the meteorological information from both sites, it was observed that the highest contributions from this factor were associated with higher wind speeds coming from the North West (NW). The seasonality observed in this factor is also consistent with the

variation in precipitation favoring the removal mechanism of dust in air (i.e., wet deposition). Similar contributions of dust to $PM_{10}$ (with comparable to lower mass concentrations) were reported by other studies in South America, like Sao Paulo: 25.7% (11.3 µg m$^{-3}$, Martins Pereira et al. 2017), Bogotá: 30% (11.2 µg m$^{-3}$, Ramírez, et al 2018[6]), or Quito: 19-21% (4.8-5.3 µg m$^{-3}$, Zalakeviciute et al. 2020).

### 3.2.2 Primary biogenic aerosol (PBA)

The Primary biogenic aerosol (PBA) factor is linked to polyols concentrations. These compounds are well known tracers for soil and fungi activity, and plants debris (Elbert et al., 2007; Samaké, et al., 2019a,b). The following most important contributors to this factor, with narrow confidence intervals are OC, K$^+$ and heavier alkanes, species that have been observed accompanying this source in other similar studies (Borlaza, et al., 2021; Chevrier, 2016). Contributing to 7 and 13% (1.5±1.0 µg m$^{-3}$ and 2.8±1.8 µg m$^{-3}$) in average to the annual $PM_{10}$ observed in El Alto and La Paz, respectively, its share increased up

to 11 and 17% (2.1±1.1 µg m$^{-3}$ and 3.4±1.7 µgm$^{-3}$) of the mass concentrations during early autumn (March-April). Minimum concentrations were observed during winter. Chevrier (2016) and Samaké (2019b) found similar results in France observing maximum concentrations of primary biogenic tracers between late spring and early autumn. Highest contribution of this factor was observed in late summer (February) in La Paz 4.4±2.4 µg m$^{-3}$, becoming the second largest source in terms of mass during this month (28%). However, it should be noted that there are fewer number of samples collected in the other summer months.

Higher contributions of this factor were consistently observed in LP compared to EA, most likely due to its closer proximity to vegetation (both, local and in the valleys to the East).

---

[6] Concentrations reported in standard conditions of temperature and pressure



### 3.2.3 MSA rich

This factor is almost entirely identified by MSA, and responsible for 100% of the MSA present in the samples. A very small fraction of OC, V, Mn, Zn, and some heavy alkanes are also present in this factor possibly hinting a small mixing with some

anthropogenic sources. It contributes to 7% ($2.0\pm0.9$ µg m$^{-3}$ and $2.0\pm1.4$ µg m$^{-3}$) of the PM$_{10}$ mass observed in El Alto and La Paz. MSA is known to result from the oxidation of the primary emissions of dimethylsulfide (DMS) typically produced by marine phytoplankton, however studies have shown other possible sources of DMS as terrestrial biogenic sources, forest biota or lacustrine phytoplankton (Du et al., 2017; Ganor et al., 2000; Jardine et al., 2015; Saltzman et al., 1983). No clear seasonality was observed, except for the slight decrease in concentrations in the months of March and October.

Neither back trajectory analysis nor association with local wind direction were useful to elucidate on the specific origin of this factor. However, Aliaga (2021) showed that air masses passing by the Titicaca Lake formed part of the third main air mass pathway arriving to the nearest GAW station (CHC-GAW) between December 2017 and May 2018. Moreover, Scholz (2022) showed that the observed DMS in CHC-GAW during the same period was mostly linked to long-range transport of marine air masses, with a smaller contribution from the Titicaca Lake. Given that air masses coming from the coast do not represent an

important source of PM in the conurbation, terrestrial or lacustrine sources could be more likely the origin of this factor. The Titicaca Lake, the largest freshwater lake in South America, is located about 50 kilometers outside the metropolitan area (about 50 km) and long-range transport of air masses from the Amazon can also be observed at our two sites.

### 3.2.4 Secondary sulfate

This factor contributes to 8% of the overall observed mass concentrations at both sites ($1.9\pm2.1$ µgm$^{-3}$ and $2.0\pm2.1$ µg m$^{-3}$ in

El Alto and La Paz, respectively) and is characterized by the presence of sulfate and ammonium. This factor is generally associated to long range transport of air masses in preceding European studies (Fulvio Amato et al., 2016; Borlaza et al., 2021; Waked et al., 2014) due to the time scales and conditions necessary to form ammonium sulfate from its gaseous precursors: sulfuric acid (H$_2$SO$_4$) and ammonia (NH$_3$) (Viana et al., 2008). It can be seen that a small fraction of other inorganic elements such as Na, K, Mg, Ca are also found with tight confidence intervals in this factor. These elements have also been observed to

be associated to sulfate rich factors in previous European studies, at times associated to long-range transport factors (aged sea salt) (Borlaza et al., 2021; Dai et al., 2020; Veld et al., 2021; Weber et al., 2019). Nevertheless, the small contribution of Zn and some heavy alkanes in the factor shows there could also be an influence of local sources to this factor. This could be attributed to the lose regulations of sulfur concentrations in imported fuels (<5000 ppm for diesel and <500 ppm for gasoline, Decree 1499/2013 of the Bolivian government), which represents 41 to 46% of the national consumed fuel (Correo del Sur,

2022). Further, this factor also includes a small fraction of OC, that could arise either from anthropogenic emissions or from biogenic SOA formation (Borlaza et al., 2021).

The highest contributions from this factor were observed during October and November (local spring) where a good combination of the key ingredients for ammonium sulfate formation is achieved: strong solar radiation, moderate temperature





and relative humidity  (Karamchandani & Seigneur, 1999; Korhonen et al., 1999).  A similar temporal variability was observed
in the city of Arequipa (Peru) (Olson et al., 2021), the closest urban high-altitude large agglomeration (ca. 2300 m a.s.l., ~1
million inhabitants[7]) 300 km to the west of LP-EA. The aforementioned study found urban combustion emissions to be the
main sources of sulfate aerosols in the city (50%), followed by dust (20%), despite its proximity to the coast and to the Central
Andes volcanic region. However, it is important to highlight that an increase in sulfur concentrations associated to an increase
in the regional volcanism activity took place during the same period (Manrique et al., 2018; Masías et al., 2016), which could
play a role in the observed seasonality. Nonetheless, the fact that the average contributions of this factor to total $PM_{10}$ is
basically the same for both cities points to an even distribution of this factor throughout the metropolitan region. Although the
overall contribution of this factor to total PM is relatively low compared to other factors, it is responsible for 14-15% of the
observed mass in both sites during spring, while only 3-4% of the total mass during winter.

### 3.2.5 Secondary nitrate

This factor is responsible of 53% of the nitrate found in the samples and represents the second largest source for the ammonium
found at both sites (23%). This factor also presents a secondary contribution with a narrow interval of confidence for EC, OC,
Zn, Pb, and heavy alkanes, tracers of traffic emissions. This evidences that the main source of the nitrates observed in La Paz
and El Alto is linked to the combustion of fossil fuels, and is mostly locally produced from the oxidation of $NO_x$ emitted from
traffic. Previous studies of emission inventories in the country also estimated mobile (transportation-related) sources to be the
main source of $NO_x$ (Herbst, 2007; Pareja et al., 2011). The contribution of this factor to total $PM_{10}$ was of 9 and 6% (2.3±2.0
µg m$^{-3}$ and 1.6±1.6 µg m$^{-3}$) in El Alto and La Paz, respectively. Larger concentrations are observed in El Alto compared to La
Paz. Since $NO_x$ concentrations were not monitored at either of the stations, we can only speculate that the difference between
La Paz and El Alto is partly due to ambient temperature difference between both cities, given that colder temperatures favor
partitioning of nitrate in particulate phase.

### 3.2.6 Biomass combustion

The main source of for biomass burning pollution in the tropical South America are agricultural practices and land use change
(Mataveli et al., 2021). Even if it is not a common practice in the Andean region, long-range transport of air masses coming
from the Bolivian lowlands and neighboring countries produces this factor at both sites with a significant contribution to PM.
The main species represented in this factor are OC, levoglucosan, mannosan, and $K^+$, which are typical tracers of biomass
burning (Li et al., 2021; Simoneit, 2002; Simoneit & Elias, 2000). Although 100% of mannosan is explained by this factor,
only about 76% of the levoglucosan present in the samples can be explained by this source (despite the applied constraint).
Low contributions of EC to this factor produce a median OC/EC ratio of 17.8. This factor contributes to 9% and 8% of the
$PM_{10}$ concentrations annually in La Paz and El Alto, with maximum average contributions of 17% and 13% (6.4±5.4 µg m$^{-3}$

---

[7] https://m.inei.gob.pe/prensa/noticias/arequipa-alberga-a-1-millon-316-mil-habitantes-9903/



and 5.4±4.7 µg m$^{-3}$) in the middle of the dry season (July-September), peaking in August. In contrast, the values during autumn
are much lower (1.0±1.0 µg m$^{-3}$ and 1.3±0.9 µg m$^{-3}$). The median levoglucosan to mannosan ratios (Lev/Man=9.1) of this
profile were found to be close to previously reported by laboratory and field studies (Hall et al. 2012: 10; Martins Pereira et
al. 2017: 11; Pereira et al. 2017: 12). The difference between cities in the observed concentrations assigned to this factor during
the biomass burning season might be explained by the fact that EA, located higher up on Altiplano, is potentially less influenced
by long range transport from the low lands.

Even if agricultural biomass burning practiced in the Andean valleys and the Amazon region of Bolivia and neighboring
countries has a relatively low contribution on annual basis, it is important during the dry season. Over the days where PM$_{10}$
concentrations exceeded the short exposure AQG recommended by the WHO (45 µg m$^{-3}$ in 24-hr), the biomass burning factor
was responsible for 13% of the total mass in EA (7.0±5.9 µg m$^{-3}$) and 23% in LP (11.9±7.4 µg m$^{-3}$. This makes biomass
burning the second most important source of PM after dust during those episodes.

### 3.2.7 Non-exhaust vehicular emissions

This factor is identified by metals such as Cu, Sn, Sn and Pb, and a significant contribution of Fe in terms of mass. These
species have been previously reported as tracers for break and tire wears (F. Amato et al., 2011; Charron et al., 2019; Fukuzaki
et al., 1986), generated by vehicles through mechanical abrasion. This factor appeared at an early stage in the single site PMF
in El Alto but it was not observable in La Paz. The multisite PMF allowed to clearly identify this factor in La Paz, splitting it
from another traffic related source. This factor contributes to 3% of the total PM$_{10}$ mass at both sites, with slightly higher
contributions during the dry season, following a similar seasonality as the dust factor. However, this factor frequently presents
high concentration spikes in El Alto that are not observed in La Paz.

### 3.2.8 Open waste burning

Thanks to the addition of PAHs and alkanes into the PMF analysis, a specific factor was identified, tentatively ascribed to
waste burning. This factor is characterized by the presence of levoglucosan, K$^+$, EC, OC, metal species such as Al, Ti, V, Rb,
Pb, PAHs and alkanes, being accountable for 57% of the Triphenylene observed in the samples. This factor also contributes
in median to 10-20% of the observed concentrations of PAH_1, PAH_2, BaA and Chr, and 15 to 35% of the measured alkanes,
and represents the second major source of the observed alkanes. Although Cl$^-$ was not included in the final PMF solution
because of the instability it added to all the solutions explored, it was observed that a significant fraction of total Cl$^-$ appears
in this factor in the preliminary runs. All these elements are typical byproducts of the combustion of plastic mixed with
vegetation or wood (Cash et al., 2021; Christian et al., 2010; Guttikunda et al., 2013, 2019; Kumar et al., 2018; Lanz et al.,
2008; Rivellini et al., 2017; Simoneit, 2002; Singh et al., 2008). Similar factors have been previously observed in prior studies
(Martins Pereira et al., 2017; Rai et al., 2020; Zíková et al., 2016), but only very few of these studies were able to distinguish
it as a separate factor from biomass burning or traffic, given the ubiquity of some of the tracers.



This factor is responsible for only 5 and 2% (1.8±1.8 µg m$^{-3}$ and 0.8±1.2 µg m$^{-3}$) on a yearly average of the total mass of PM$_{10}$ observed in El Alto and La Paz, respectively, but its contribution can increase up to 9 and 6% (3.4±1.6 µg m$^{-3}$ and 2.1±1.2 µg m$^{-3}$) during winter. The seasonality of this factor is clear with rising up share in May and declining in August. The exact source of this factor has not been identified yet, but higher contributions in El Alto than in La Paz tend to point to local sources. Analysis of winds characteristics shows that higher concentrations of this factor are linked to low winds speeds blowing from

the North in the case of El Alto, and are from the northwest and with higher wind speeds in the case of La Paz. The local emissions could come whether from punctual-source waste burning, or the emissions of industrial and open commercial areas in El Alto, being transported to the city of La Paz. Similar behavior was observed when associating Cl$^-$ to wind speed and wind direction (not presented here).

### 3.2.9 Traffic sources 1 and 2 (gasoline/diesel)

The first traffic factor resolved (TR1) is annually responsible for 6 and 8% of the observed PM mass in La Paz and El Alto, respectively (1.9±2.0 µg m$^{-3}$ and 2.3±2.0 µg m$^{-3}$). The main tracers of this factor are a small fraction of EC and OC, the presence of metals such as Na, Ca, Mg, Al, Fe, Ti, V, Mn, Zn, Rb, Pb, and over 40% of most PAH concentrations, consistent with previously observed vehicular emission factor profiles (F. Amato et al., 2011; Charron et al., 2019; Waked et al., 2014). Some traces of sulfate, and lighter alkanes can also be observed in the chemical profile of this factor. The second traffic factor

(TR2) contributed with an average of 23% and 13% to total PM$_{10}$ in La Paz and El Alto (5.7±3.5 µg m$^{-3}$ and 3.6±2.5 µg m$^{-3}$). The chemical species identified in this factor are similar to those of TR1: EC, OC, Zn, PAH_1 and Cor, with small contributions of sulfate, Na, Ca, Mg and Mn. It is noteworthy that no alkanes and almost no hopanes are contributing in TR2, even if these compounds are in principle emitted by road traffic.

  The median OC/EC ratios obtained from the traffic chemical profiles of TR1 (0.4) and TR2 (1.1). Having low OC/EC ratios

in high-altitude conditions is no surprise, since combustion processes are less efficient under low O$_2$ availability (Wang, et al., 2013a). However, because of the very different conditions for combustion, literature values of the ratio OC/EC (> 1 for gasoline, and < 1 for diesel, Y. Cheng et al., 2010; Yan Cheng et al., 2021; Ka Wong et al., 2020; H. H. Yang et al., 2019) cannot be used to identify which of the traffic factors can be associated to gasoline- or diesel-powered vehicles. An important difference between the two factors is that a ratio of Mn/Zn>1 is found in TR1, which is the opposite in the case of TR2. The

fuel analysis (pre-combustion) showed that the largest differences in the chemical composition between local gasoline and diesel fuels was the relative abundance of Mn compared to Zn. Whilst the measured prior-combustion ratios of Mn/Zn are not preserved, the Mn/Zn ration is a characteristic feature of each of the profiles. Moreover, TR1 have higher PAH concentrations, whereas TR2 showed much lower contributions of PAHs. Previous studies have shown that gasoline-powered vehicles indeed emit more long-chain PAHs than diesel fuel (IFP, 2021; Leoz-garziandia et al., 2014; Zielinska, et al., 2004a). While gasoline-

powered vehicles represent over 80% of the vehicle fleet in Bolivia, literature has shown that diesel-powered vehicles can emit 10 to 30 times more particles than gasoline (Zielinska et al., 2004b).





In terms of contribution, the overall influence of TR2 is more important than TR1 in La Paz, and is almost twice as high as the influence of TR2 in El Alto. This can be related to the difference in topography, since several studies have shown that steep slopes can significantly increase the vehicle fuel consumption (Carrese et al., 2013; Y. Wang & Boggio-marzet, 2018). Also,

the proximity of the LP sampling site to the nearest main avenue (~100 m) and to the parking lot of the municipality buses (~100 m, horizontal distance; ~45 m vertical distance), which are diesel powered vehicles, might play an important role in the respective influences of TR1 and TR2 in LP. The PD-SID comparison of both traffic factors with the road traffic profiles of several urban/urban-background French sites presented in Borlaza (2021) and Weber (2019) (SI)  showed there is a significant similarity between TR2 and the French road traffic factors (where diesel is the dominant fuel used). However, TR1 presented

PD values outside the similarity thresholds established by Pernigotti & Belis (2018).

Based on the previous description of factors TR1 and TR2, we consider that TR1 is likely related to the emissions of gasoline-powered vehicles, whereas TR2 is most likely associated to diesel-powered vehicles. However, the number of registered cars reported by the Municipal Tax Administration in 2011 showed that the number of gasoline-powered vehicles in the city of La Paz (~90% of the registered vehicle fleet in La Paz) was 2.4 times larger than the ones registered in El Alto (~80% of the

registered vehicle fleet in El Alto). In contrast, similar number of diesel-powered vehicles were registered at both sites. If these numbers would be directly related to the flow of vehicles in the metropolitan area, they could imply the opposite of what can be concluded from the chemical profiles, i.e. TR1 associated to diesel powered-vehicles and TR2 associated to gasoline-powered vehicles. However, it should be kept in mind that registration in one city does not mean that the vehicle circulates in that location. This could be especially the case for trucks and buses that move between La Paz and El Alto. In addition, it is

known that large contributions of emissions could come from a small number of vehicles (Ježek et al., 2015). All these factors make difficult to estimate the contribution of the different type of vehicles circulating in the metropolitan area to the measurements taken on the filters.

The ensemble of both factors constitutes the major source of particulate matter in the case of La Paz, and the second largest source of $PM_{10}$ particles in the case of El Alto. TR1 displays a slight seasonality displaying higher concentrations during the

dry season of 2016. In the case of TR2, not much seasonality is observed except for the higher concentrations observed between April-May 2016 and May-June 2017. Although, one could expect to have similar variability for traffic-related profiles, this is not the first study to observe a difference in the yearly variability between gasoline and diesel factors (Squizzato et al., 2018, for a study in New York State).

### 3.2.10 Lubricant oil

The addition of molecular organic species (PAH, alkanes, and hopanes) allowed the identification of a factor attributable to lubricant combustion, likely associated to vehicle emissions. This factor is marked by the presence of hopanes and alkanes, some of them being univocal tracers of oil combustion (Charron et al., 2019; El Haddad et al., 2009). It is responsible of 36-47% of the mass of alkanes present in the samples, and the major source of the total mass of hopanes (65%). This factor also presents smaller percentage contributions of OC, $K^+$, Na, Ca, V, Mn, Cu, Zn, and some PAHs, elements also present in fuel



combustion emissions. The contribution of this source to annual $PM_{10}$ mass is of 3 and 1% (0.9±0.8 µg m$^{-3}$ and 0.4±0.6 µg m$^{-3}$) in La Paz and El Alto, respectively. A clear increase in contributions during the coldest months of the year can be observed in the variability of this factor. A similar evolution of the hopanes with maximum concentrations during winter was observed in Marnaz (France) by Chevrier (2016). Likewise, a study in three cities of the United states observed an increase of concentrations of hopanes and alkanes during the coldest months of the year (Kioumourtzoglou et al., 2013). This seasonality

could be associated to the cold start of vehicle engines during the early morning and late-night hours, in the period when minimum temperatures drop.

In general, the contribution of this factor in the city of La Paz is higher than in El Alto. This can be associated to the stress of vehicle engines when driving through the steep streets of La Paz, effort that is minimized in El Alto due to its flat topography. Even if the contributions of this factor to total $PM_{10}$ mass are relatively low, it becomes important in terms of air quality since

it is one of the major sources of alkanes and hopanes, the latter being considered hazardous for human health since it has proven to be associated to systemic inflammation biomarkers (Delfino et al., 2010).

### 3.3 Methodology discussions

The sampling strategy, the complete chemical characterization, and the multisite PMF, coupled with the specific geographical patterns, permitted this quite unique study to offer an extensive characterization of PM sources in high-altitude cities and

should provide important information that can help policy-making towards better air quality in the region. However, we are aware of some limitations.

- PMF limitations

Having enough samples in the multisite approach and a fairly large chemical speciation including organic tracers allowed to resolve 11 factors in the PMF analysis. It is worth noting that only few studies have been able to resolve similar number of

sources with good statistical indicators (Borlaza et al., 2021; Chevrier, 2016; Pandolfi et al., 2020; Waked et al., 2014; Weber et al., 2019). However, when attempting a larger number of factors generates instability in the otherwise geochemically stable profiles. There can be several factors contributing to this observed limitation, among which can be found:

- The collinearity between sources, creating mixed factors. The presence of OC in both secondary sulfate and primary biogenic emissions could speak of a possible mixing of these factors with biogenic secondary organic aerosols

(BSOA). A detachment of BSOA was not possible due to the lack of the specific tracers of this source (3-MBTCA, or cellulose, or methyltetrols).

- Even though industry is not fully developed, there are factories within and the surroundings of the Metropolitan area that were not resolved by the PMF (e.g. cement plants, brickyards, PVC manufactory plants). This could be due to

the lack of specific tracers for these sources in the analysis, a similarity of the chemical profile and temporal variability of the emissions compared to the resolved sources, or simply because they represent a very small fraction of $PM_{10}$





- The need for removing chloride from the analysis for bringing instability to the solution. This was likely associated to the large variability of this volatile compound.


• Multisite approach limitations

Although the use of the multisite approach added value to the results obtained with the single-site approach, it is important to note that *a priori* this method cannot be applied to sites that differ greatly from each other. It was important to verify the similarity of the single-site solutions. However, a drawback of the multisite approach is that it will force the similarity of the common factors found between the two sites, smoothing out the specificity of them. Some examples of this forced similarity are listed below:

- The multisite approach allowed the separation of EC (a traffic tracer) out of the dust profile successfully. However, considering that the Altiplano is a major source of dust and that the only path that the air masses take when transporting dust from the Altiplano to La Paz is across both cities, it is not surprising that the dust factor in the city of La Paz (single-site solution) is highly influenced by traffic tracers. For the multisite solution the indirect information of the mixing of sources during transport is lost.
- The molar ratio of sulfate and ammonia concentrations in each of the cities gives different average values (2.05 and 1.63 in El Alto and La Paz, respectively), which provides information that in the city of La Paz there is less available ammonium to neutralize sulfate and nitrate ions. However, this is no longer observable in the multisite analysis, in which a median molar ratio of 1.96 is found representative of both cities.
- The MSA-rich profile shows in El Alto strong mixing with metallic species, among them crustal material, which hinted its path through the Altiplano towards the city of El Alto. This in exchange is no longer seen after the multisite analysis.

Given that the benefits of a more specific characterization of the sources thanks to the multisite approach outweigh the drawbacks of using it, we consider that this was the best technique to be applied in the metropolitan region of La Paz and El Alto with such database in hand.

## 4 Conclusions

This study brings innovative information and a unique analysis of air pollution sources in the high-altitude urban environment of the fast-growing cities of La Paz and El Alto in Bolivia. It also provides a detailed description of the chemical profiles of 11 identified source types resolved by the multisite PMF method and, their temporal and spatial variability. The wide and comprehensive dataset and the combination of inorganics and organics species allows an advanced source apportionment going beyond classical solutions, allowing the identification of several biogenic factors and combustion-related factors that otherwise



would have gone unresolved. For instance, waste burning was separated from biomass burning, and traffic exhaust emissions were separated into two independent profiles.

On average, vehicular emissions represent 35 and 25% of the $PM_{10}$ concentrations measured in La Paz and El Alto. Then, dust stands out as one of the two main sources contributing to 20 and 32%. The factors associated with secondary inorganic aerosols account for 22 and 24% and the primary biogenic emissions account for 7 and 13% at the annual level. One of the smallest factors in terms of contribution to the total mass but the second most important factor responsible for the observed PAHs is the non-regulated burning of waste happening mostly in El Alto between May and August.

The observations in this study are from urban background sites, representing wider region pollution levels in La Paz and El Alto. Locally, especially near roads or landfills, the mass concentrations are expected to be higher. Even if most of the resolved sources are associated with local activities (dust resuspension, primary and secondary vehicular emissions, and waste burning), there is a significant contribution of regional natural anthropogenic sources of PM (Primary and secondary biogenic emissions, and biomass burning).

Based on our results, we can outline relevant actions towards improvement of air quality in La Paz and El Alto:

1) Regulation of Vehicular emissions have to improved. As the Metropolitan area continues to grow, more efficient means of transportation and stricter policies and control on combustion practices are needed to ensure that air quality is not further degraded.

2) Waste burning should be prohibited. It is a major source of PAHs and other pollutants with high human health risk factor.

3) Agricultural biomass burning is a seasonal source, a decrease in their emissions would result in a significant improvement in the air quality during the most polluted season, not only for the metropolis but also for the rest of the country.

4) Dust is important source in terms of mass and it also has anthropogenic component in it (e.g. vehicle resuspension, construction activities, mining).

5) Updated policies of pollutant emissions are essential to regulate also the growing industry sector.

In order to have a comprehensive understanding of the pollution sources in the metropolitan area of La Paz and El Alto, information on the gaseous components is of utmost importance. A longer sampling time period together with an updated emissions inventory of the resolved sources would be beneficial for a better understanding of the resolved sources and their evolution in time. Furthermore, analyzing the potential impact on health of the resolved sources is crucial for efficiently targeting the most hazardous sources of PM.

**Code availability.**

The software code is available upon request.

**Data availability.**

The chemical and PMF datasets are available upon request.

**Authors contribution.**

GU, MA, PL, JLJ, AA, JLB, RK, IM, NP and AW participated in the conceptualization of the experimental set up and design. IM participated in the data curation. VM, MP and LJB participated in the formal analysis and the development of the methodology. GU, MA, PL, JLJ, AA, JLB, RK, PG were involved in the funding and resource acquisition. JLB, IM, NP and VC contributed to the investigation by organizing the samples collection and performing the experiments. GU, MA, PL, MP, LJB, GM and JLJ helped with mentoring, supervision and validation of the methodology, techniques and results. VM was responsible for the data processing and the writing of the original draft. GU, PL and JLB revised the original draft. All the authors reviewed and edited the manuscript.

**Competing interests.**

G. Močnik is employed by Haze Instruments d.o.o., the manufacturer of the aerosol instrumentation.

**Acknowledgements.**

Authors wish to thank all the many people from the different laboratories (LFA, Idaea-CSIC, IGE, Air O Sol analytical platform, EDYTEM) who actively contributed over the years in filter sampling and/or analysis. Specifically, thanks to Samuel Weber, Federico Bianchi, Claudia Mohr and Diego Aliaga for the active participation in the discussions of the obtained results; J.C. Franconny and M. Pin who carried out the organic compounds analysis by GC-MS on the PTAL analytical platform of EDYTEM; the engineers F. Masson, F. Donaz, C. Vérin, A Vella, R El Azzouzi and many technicians who performed ECOC, ionic chromatography and HPLC-PAD on the Air-O Sol plateform; S. Rios and E. Miranda of GAMLP (Gobierno Autónomo Municipal de La Paz) who provided access and facilitated tasks at Pipiripi; IIF personnel that helped in logistics during the campaign; Undergrad students who collected samples: Y. Laura, G. Salvatierra, M. Roca, D. Calasich, E. Huanca, Z. Tuco, S. Herrera, M. Vicente, M. Zapata, R. Copa.

**Financial support.**

This research has been supported by the Institute de Recherche pour le Développement (IRD) France and IRD delegation in Bolivia, Javna Agencija za Raziskovalno Dejavnost RS (grant nos. P1-0385), Grant Agency of the Czech Republic 19-15405S.



The Labex OSUG@2020 (ANR10 LABX56) provided some financial support for instruments on the Air O Sol analytical platform, EU H2020 MSCA-RISE project PAPILA (Grant #: 777544).

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

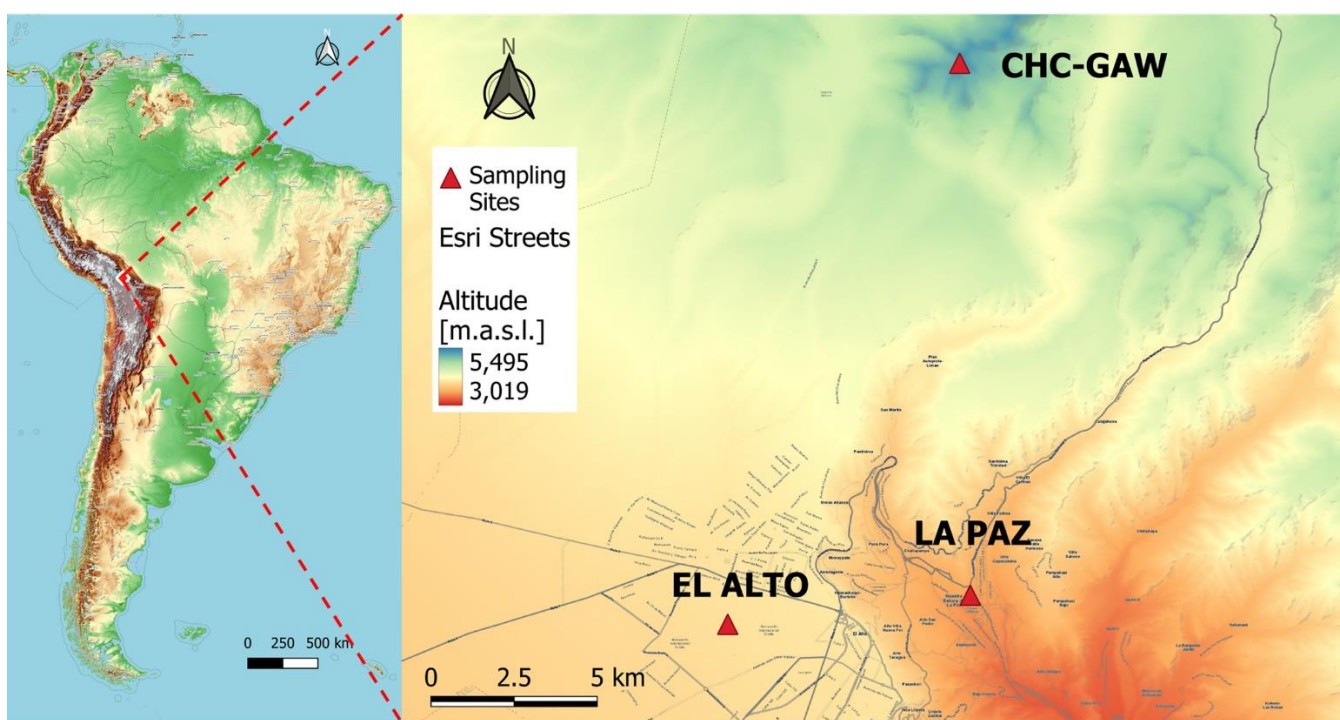

**Figure 1. Geographical location of the sampling sites (left panel) La Paz (LP) and El Alto (EA) zoomed in (right panel) and positioned with respect to the regional Chacaltaya-GAW monitoring station (CHC-GAW). Color scale represents the altitude above sea level (ESRI Streets 2022).**





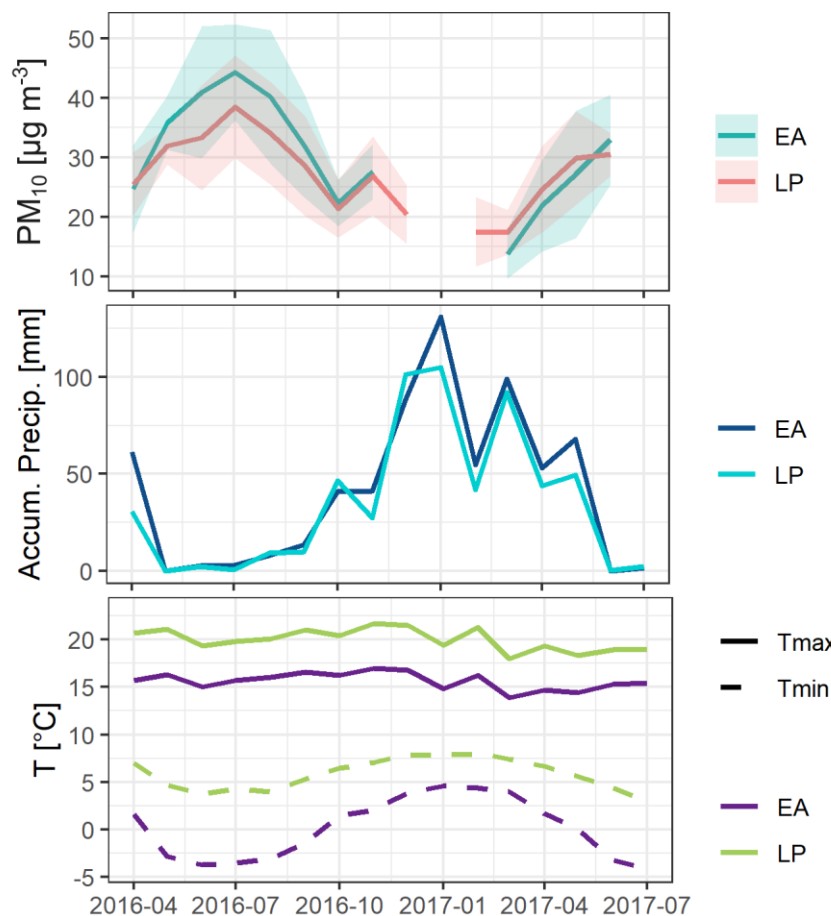

Figure 2. Monthly PM10 mean concentrations (µg m-3), monthly accumulated precipitation (Accum. Precip., mm), and monthly mean maximum/minimum temperature (°C).





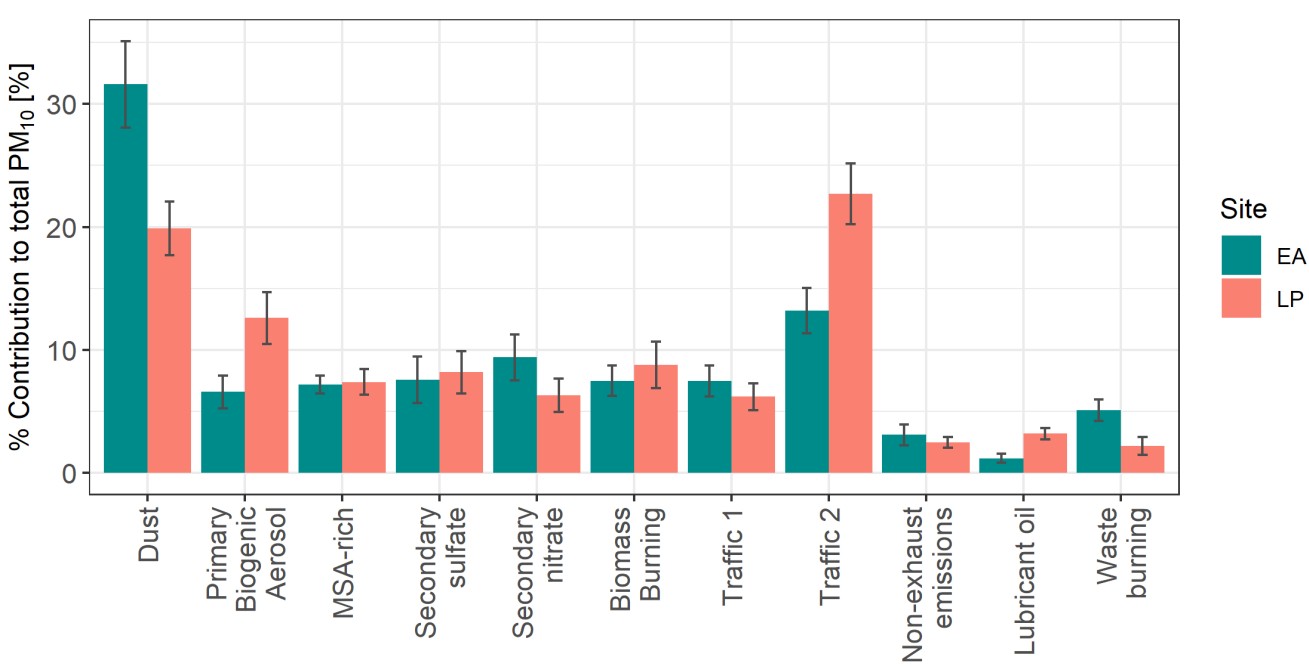

1030

**Figure 3.** Average factor contributions to total PM10 at each site, resulting from the multisite PMF. The bars represent the 95% confidence interval of the mean values.







**Figure 4. Source mass-contribution monthly variations (n = number of modeled data points included in the average) between April 2016 and July 2017.**



**Figure 5. Source chemical profiles (bars representing median bootstrap mass contributions of each specie per µg of PM mass attributed to each source in y-axis, red dots represent mean DISP values, error bars represent DISP confidence intervals, color scale represent the contribution in percentage). The name of each source is further described and developed in the individual factor descriptions[8].**

---

[8] PAH_1: [BghiP]+[IP]+[BbF]; PAH_2: [Fla]+[Pyr]).



**Table 1. Set of constraints applied to final solution**

| FACTOR | SPECIE | CONSTRAINT | VALUE |
|---|---|---|---|
| **Biomass Burning** | Levoglucosan | Pull up maximally | %dQ 0.50 |
| **Biomass Burning** | Mannosan | Pull up maximally | %dQ 0.50 |
| **Primary Biogenic Aerosol** | Polyols | Pull up maximally | %dQ 0.50 |
| **MSA-Rich** | MSA | Pull up maximally | %dQ 0.50 |

**Table 2. Air quality studies at high-altitude Latin American cities.**

| | Average $PM_{10}$ (Min-Max) [µg m$^{-3}$] | Period | Study | Population[9] | Altitude [m a.s.l.] |
|---|---|---|---|---|---|
| **Mexico City, Mexico** | (51-132) | March, 2006 | (Mugica et al., 2009) | 18,457,000 | 2,850 |
| | (19-174) | Jul-Dec, 2000 | (Gutiérrez-Castillo et al., 2005) | 19,444,000 | |
| **Quito, Ecuador** | 24.9-26.2 | Jan-Oct, 2017 | (Zalakeviciute, et al., 2020) | 1,793,000 | 2,240 |
| **Bogota, Colombia** | 37.5 (9.9-160)[10] | Jun, 2015-May 2016 | (Ramírez, et al., 2018) | 9,989,000 | 2,620 |
| **El Alto, Bolivia** | 29.9 (6.6-59.0)[11] | April 2016-June 2017 | Present study | | 4050 |
| **La Paz, Bolivia** | 27.2(11.6-50.9)[6] | April 2016-June 2017 | Present study | | 3200-3600 |

[9] https://populationstat.com/
[10] Concentrations reported in standard conditions of temperature and pressure.
[11] Campaign average $PM_{10}$ concentrations that could slightly over estimate annual mean values due to a low number of samples collected during the wet season, where the minimum mass concentrations expected.