# Peer review of "Source apportionment study on particulate air pollution in two highaltitude Bolivian cities: La Paz and El Alto"

_Atmospheric Chemistry and Physics, 2022_

## Author Comment (AC1)

The authors would like to thank the anonymous referee # 1 for taking the time to review the manuscript. We thank them for their kind and encouraging words, and for the very relevant comments that allowed us to improve the manuscript. We followed their advice, and an additional grammar revision was made to the original text.

Below you will find the list of the referees' observations (bold), right after, each of the author's responses (normal font) and the respective changes made to the manuscript (italic), highlighting the sections that were modified.

1. **Lines 31-32 - Some studies as Serafeim et al., 2023 (Atmospheric Environment v. 298, 119593) do include PAHs, and WSOC together with inorganic species for São Paulo (Brazil) particulate matter, however, this may be the first including hopanes and alkanes.**
   As pointed out by the referee, the publication of recent papers during the last few months led us to update the phrase to:
   *"This is one of the first source apportionment studies in South America incorporating a large set of organic markers (such as levoglucosan, polycyclic aromatic hydrocarbons - PAH, hopanes and alkanes) together with inorganic species."*

2. **Line 183 - It is always important to mention that non-sea salt potassium is not exclusively emitted by biomass burning. This species may be also very linked with soil resuspension and is also related to fertilizers (Urban et al., 2012 - Atmospheric Environment, v. 61, p. 562-569), especially in dusty sites such as El Alto (which depends on the local soil composition, of course).**
   We agree with the remark made by the referee, for which the sentence was modified to:
   *"…only the ICP-MS metals were included in order to avoid double counting, except for $K^+$, for which the IC measurements were used since water soluble $K^+$ is a known tracer for biomass burning (BB), soil resuspension, and fertilizers (Li et al., 2021; Urban et al., 2012)."*

3. **Lines 271-276 - Are these studies comparable? Were they also performed year-round (or specific polluted period)? This should be discussed.**
   We updated Table 2 to only include studies that report average concentrations of PM measured during at least a 1-year period.
   We believe that comparisons with both South American data and data from European countries reporting similar PM concentrations are indeed valid and relevant. In order to compare PM concentrations at different locations, it is necessary that these concentrations are under the same conditions of pressure and temperature. Therefore, the PM concentrations measured in the present study were compared with the concentrations reported in the high-altitude Latin American cities and the mentioned European cities under standard conditions of pressure and temperature (STP). The only exception was the study conducted in Mexico City, where there is no mention of the concentrations being transformed to STP conditions, for which it is assumed that they are reported in ambient conditions. However, it can be observed that even the ambient concentrations (which are lower than the STP concentrations) are similar or higher than the STP concentrations of the present study. We can, thus, conclude that the latter are indeed lower. It should be noted that this paragraph only discusses the absolute PM mass concentrations and not the exposure to PM of the inhabitants of each city. We believe that comparing the concentrations found in the present study with other cities with similar topographic and pollution characteristics puts our results in context.

   *"The observed concentrations are lower compared to those reported for Mexico City, a high-altitude (2850 m a.s.l) Latin-American megacity (Table 2), but higher than those observed in the*

*cities of Bogotá and Quito. The average concentrations found in La Paz-El Alto are nearly double the reported average concentrations for most suburban and urban background sites in Europe, and similar to those measured in Turkey, certain regions in Poland (Rybnik: 44.1 µg m⁻³), Bulgaria (Vidin: 41.3 µg m⁻³), North Macedonia (Skopje: 48.7 µg m⁻³) and Italy (Napoli: 46.9 µg m⁻³) in 2019 (EEA, 2020; EEA, 2022)."*

**Table 1. Air quality studies at high-altitude Latin American cities.**

| | Average PM$_{10}$ (Min-Max) [µg m$^{-3}$] | Period | Study | Population[i] | Altitude [m a.s.l.] |
|---|---|---|---|---|---|
| **Mexico City, Mexico** | (45.38-80.10) [ii] | 2015-2016 | (Cárdenas-Moreno et al., 2021) | 18,457,000 | 2,850 |
| **Quito, Ecuador** | 24.9-26.2 [iii,iv] | Jan 2017-Dec 2018 | (Zalakeviciute, et al., 2020) | 1,793,000 | 2,240 |
| **Bogota, Colombia** | 37.5 (9.89-160) [iii, iv] | Jun, 2015-May 2016 | (Ramírez, et al., 2018) | 9,989,000 | 2,620 |
| **El Alto, Bolivia** | 29.9 (6.6-59.0) [iii,v] | April 2016-June 2017 | Present study | | 4050 |
| **La Paz, Bolivia** | 27.2 (11.6-50.9) [iii, v] | April 2016-June 2017 | Present study | | 3200-3600 |

[i] https://populationstat.com/
[ii] Range of spatial variation
[iii] Range of seasonal variation
[iv] Concentrations reported in standard conditions of temperature and pressure
[v] Campaign average PM10 concentrations that could slightly over estimate annual mean values due to a low number of samples collected during the wet season, where the minimum mass concentrations expected.

4. **Lines 280-284 - Some authors attribute those differences to adsorbed water in the aerosol and to the presence of non-measured species (i.e. carbonates), which depends on the way PM was reconstructed (Pio et al., 2013 - Atmospheric Environment, v. 71, p 15-25)**

   We agree with the relevant remark; therefore, we included this alternative causes of the observed difference between measured and reconstructed PM.

   *"Average PM$_{10}$ (recons.)/ PM$_{10}$ (meas.) ratios of 0.91 in El Alto and 0.82 in La Paz were found. The remaining unidentified mass fraction may be attributed to the loss of volatile organic matter and secondary aerosols post-weighing, during the transport of the filter fractions to be analyzed. The difference can also be associated to the presence of non-measured species (i.e. carbonates) or to the adsorption of water in the aerosol particles or the filter (Pio et al., 2013). Moreover A 10% uncertainty associated with the gravimetry measurements could also have a role in the observed difference".*

5. **Section 3.2 - The authors may consider using Latin American review studies for the source apportionment interpretation and discussions, such as La Colla et al., (2020 - Environmental Reviews v. 29, n. 3), or emission studies such as Brito et al (2013 - ACP, v. 13, p.12199-12213) and more recently Pereira et al. (2023; v. 856, 159006) performed for the biofuel-impacted fleet in São Paulo (Brazil), they can be helpful. The use of biofuels in Bolivia could be discussed, as well.**

   We thank the referee for the suggestion of the mentioned papers to enrich the discussion of the profiles obtained in the PMF. The first review compiles much information on the chemical speciation of the metalloids found in five South American megacities and their associated sources. The second one gives

a detailed description of the finger print of the different vehicular emissions in a tunnel experiment in Sao Paulo, Brazil. Unfortunately, the third one was not open access. Although the information found in the mentioned articles is invaluable, the arguments found useful to broaden up the discussion did not provide more clarity on the identification or the description of the resolved sources, and most of them had been already included in the discussion of each of the source descriptions. Moreover, the abundance of oxygen at different altitudes plays a key role in combustion processes, and the description of the emissions measured at sea level are not necessarily preserved at high altitude. Nevertheless, we will include these references as part of the references cited in the corresponding sections to reinforce the discussion.

*"This factor is identified by the presence of metals such as Cu, Sn, Sb, and Pb, along with a significant contribution of Fe in terms of mass. These species have been previously identified as tracers for break and tire wears (Amato et al., 2011; Charron et al., 2019; Fukuzaki et al., 1986), generated by vehicles through mechanical abrasion. However, some studies have also found these tracers to be associated with industrial emissions (La Colla et al., 2021), for which we could not entirely neglect the possibility of having an influence of industrial emissions masked within this factor".*

*"However, because of the very different conditions for combustion, literature values of the ratio OC/EC (> 1 for gasoline, and < 1 for diesel, Brito et al. (2013); Cheng et al. (2010); Cheng et al., (2021); Wong et al. (2020); Yang et al. (2019)"*

*"This could be especially the case for trucks and buses that move between La Paz and El Alto. In addition, it is known that large contributions of emissions could come from a small number of vehicles (Brito et al., 2013; Ježek et al., 2015; La Colla et al., 2021)."*

During the time of the experiment biofuel was not available as a vehicle fuel. We acknowledge that the inclusion of biofuel in the market after 2017 and its impact on the air quality of the conurbation remains to be studied. Likewise, the cultivation of the natural resources to produce biofuel could potentially modify the biomass burning emissions during the biomass burning season.

A detailed analysis of the typical ratios of the different PAHs as a way to distinguish the different combustion sources was not approached in the present study and remains to be presented in a future scientific article.

6. **Lines 350-354 - Were these mass concentration values given by those authors or calculated here for this manuscript? If these values were calculated by you, this should be mentioned somewhere in the manuscript.**
A remark was added in the text in the form of footnote to clarify that the absolute contributions of dust in the mentioned studies was calculated from the percentages reported in the different studies.

*"Similar contributions of dust to $PM_{10}$ (with comparable or lower mass concentrations) have been reported by other studies in South America, like Sao Paulo: 25.7% (11.3 $\mu g\ m^{-3}$, Martins Pereira et al. 2017), Bogotá: 30% (11.2 $\mu g\ m^{-3}$ (STP), Ramírez, et al 2018), and Quito: 19-21% (4.8-5.3 $\mu g\ m^{-3}$, Zalakeviciute et al. 2020) (Absolute mass concentrations of dust [$\mu g\ m^{-3}$] were calculated based on the percentage contributions reported on the studies mentioned and the reported average PM - mass concentrations)."*

7. **Lines 430-432 - These Lev/Man ratios were related to sugarcane burning, some ratios can be associated with specific types of biomass. That should be considered in the discussions. Check the excerpt from Zhang et al. (2015 - Atmospheric Environment v. 102, p. 290-301): "[...] An**

overview of biomass burning tracer ratios derived from various source profiles is provided in Table 4. In general, low LG/MN ratios (typically 3–7) are generated by softwood combustion. In contrast, burning of hardwoods and crop residues is typically associated with higher LG/MN ratios with a rather broad range of 10–83. Similar characteristics were also observed from the burning experiments of typical biofuels in South China.[...]"

A specification was added to the text:

*The median levoglucosan to mannosan ratios (Lev/Man=9.1) of this profile were found to be close to ratios previously reported for sugarcane burning (one of the main plantations in the Brazilian Amazon region) in laboratory and field studies (Hall et al. 2012: 10; Martins Pereira et al. 2017: 11; Pereira et al. 2017: 12; Zhang et al., 2015).*

8. **Line 432 - The contribution difference seems to be quite small (8 and 9 %).**

   We agree with the referee, for which we emphasized the similarity of the annual contributions in the text:

   *"This factor has similar annual contributions of 9% and 8% to $PM_{10}$ concentrations in La Paz and El Alto, with maximum average contributions of 17% and 13% (6.4±5.4 µg m$^{-3}$ and 5.4±4.7 µg m$^{-3}$) in the middle of the dry season (July-September) …"*

9. **Line 441 - Sn is duplicated here.**

   We thank the referee for spotting this typing error. The duplicated Sn was replaced by Sb.

   **Cu was associated with gasohol and ethanol-powered vehicles' exhaust in Brazil (which may also be related to the corrosion of engine parts with the increase in the use of ethanol fuel) by Brito et al (2013) and Pereira et al., (2023).**

   We thank the reviewer for taking the time to provide us with interesting points of discussion specific to the region. In the case of the of our Non-Exhaust factor, we had only considered brake abrasion as the main source of Cu because Cu was not present in the fuel analyzed, and because during the time of the experiment biofuels were not available as vehicle fuel yet.

10. **Line 453 - The authors could perform correlations of PMF factor contributions and chloride concentrations to confirm that.**

    We thank the reviewer for the suggestion, the following table was added to the SI to clarify the argument.

    Table S6. Spearman correlations between chloride and each of the resolved sources of PM. Strongest correlations are found between Cl- and Waste burning, secondly with TR1 and Non-exhaust.

| | | Waste burning | Sec. sulfate | TR1 | MSA-rich | Lubricant | BB | Dust | Sec. Nitrate | Non-Exhaust | PBA | TR2 |
|---|---|---|---|---|---|---|---|---|---|---|---|---|
| Spearman | Cl- EA | 0.75 | -0.22 | 0.57 | 0.34 | 0.28 | 0.47 | 0.59 | -0.24 | 0.67 | -0.19 | 0.25 |
| | Cl- LP | 0.67 | 0.01 | 0.61 | 0.39 | 0.49 | 0.53 | 0.45 | 0.19 | 0.57 | -0.25 | 0.44 |

    Reference in text:

    *"A Spearman correlation >0.67 was found between the concentrations of Cl$^-$ and the PM concentrations attributed to this factor (Table S6)"*

11. **Line 469 - Do these factors increase with the same wind direction? Could they be from vehicular sources from different locations?**

The concentrations do not increase with the same wind direction, however winds in LP are channeled up and down the canyon due to the topography. The much higher concentrations in EA, indicate the proximity of the source to the station in EA. The following image was added to the SI for clarification.

[Figure]

*Figure S7. Polar plot showing the mean concentrations attributed to open waste burning and the associated wind speed (m s⁻¹) and wind direction.*

In text modification:

*"Analysis of wind characteristics shows that higher concentrations of this factor are linked to low wind speeds blowing from the North in the case of El Alto, and from the northwest and with higher wind speeds in the case of La Paz (Figure S7). The local emissions could originate from punctual-sources of waste burning, or the emissions of industrial and open commercial areas in El Alto, later transported to the city of La Paz. Similar behavior was observed when associating Cl⁻ to wind speed and wind direction (not presented here)."*

**12. Line 598 - What do the authors mean by "natural anthropogenic sources of PM "?**

We thank the reviewer for spotting this typing error that is now corrected: "there is a significant contribution of regional natural *and* anthropogenic sources of PM (Primary and secondary biogenic emissions, and biomass burning)"

**Technical corrections:**

All the technical corrections (numbering, subscripts, superscripts, reference format, etc.) were considered and applied.

1. **Line 108 - The authors may consider rewriting this part "Although the characteristic tropical seasonal change between a dry and a wet season"**
   We agree with the referee's suggestion. The rephrased paragraph reads as follows:
   *"The meteorological conditions throughout the year are governed by the seasonal transition between a dry and a wet season, typical of tropical regions. Temperature and wind patterns vary substantially between the two cities due to the differences in altitude and local topography".*
2. **-Line 137 - PM10 ("10" should be subscript).**
   We thank the referee for the remark, all the subscripts and superscripts were checked.

3. **Line 183 - "Biomass Burning".**
   The capitalization mistake was corrected: *"biomass burning"*

4. **Lines 188, 194, 198, 202, 205, 218, 1038, and Table 1- The word "species" is the same in the singular and plural, the authors may revise that all over the text.**
   We appreciate the referees' remark; the word specie was replaced by *"species"* throughout the document.

5. **Line 251 - "Belis et al. 2019" - Check if the citation format is correct (in other parts of the text, as well).**
   The format of all citations was checked and corrected when needed, as it was the case for this in-text citation: *Belis et al. (2019)*

6. **Lines 263-269 - "10" should be subscript on PM10 and "-3" superscript on m-3.**
   All the subscripts and superscripts were checked.

7. **Lines 236-238, 271, 288, 305, 311, and 323 - Check if figure and table citations follow the journal rules, there different formats throughout the manuscript.**
   The format of all tables and figures citations was checked and corrected when needed.

8. **Lines 285-287 - Consider rewriting this more clearly.**
   We agree with the referee's suggestion. The rephrased paragraph reads as follows:
   *"The  average percentage contribution of the chemical species that significantly contribute to the measured $PM_{10}$ concentrations in El Alto  was: 22±5% OM (i.e. 1.8·OC), 5±2% EC, 9±5%  secondary inorganic aerosols ($NH_4^+, NO_3^-, and SO_4^{2-}$), and 12±3% of crustal material (Al, Fe, Ti, Ca, K, Mg, Mn, P). In La Paz, 25±5% OM, 6±2% EC, 8±5%  secondary inorganic aerosols, and 10±2% of crustal material".*

9. **Lines 352, 432-433, and 458 - Martins Pereira, 2017 and Pereira 2017 are different articles from the same author. They may be cited as "Pereira et al., 2017a" and "Pereira et al., 2017b".**
   We thank the referees' remark, the citation of these two papers was corrected throughout the manuscript.

10. **Line 360, 384... - Space between "µg" and "m-3" throughout the text.**
    The correction was applied throughout the manuscript

11. **Line 442 - "Amato et al., 2011" - Check if the citation styles are correct (in other parts of the text, as well).**
    We thank the reviewer for the remark. The format of all citations was checked and corrected when needed

12. **Lines 528 - "United States".**
    The correction was applied.

13. **References section - Check if all the references are following the ACP guide for authors. Improve text details, as well (superscript and subscript).**
    The format of all citations was checked and corrected when needed.

14. **Figure captions - Check superscript and subscript.**
    All the figure captions were checked and corrected.

---

## Author Comment (AC2)

The authors would like to thank the anonymous referee # 2 for taking the time to review the manuscript. We thank for validating our work and for providing us with valuable insights that allowed us to improve the manuscript.

Below you will find the list of the referees' observations (bold), right after, each of the author's responses (normal font) and the respective changes made to the manuscript (italic), highlighting the sections that were modified.

**Minor comments**

1. **Fig. 1. The detail map of the sampling site has wrong color scale (opposite to the map of South America) and must be corrected.**

   We thank the reviewer for this remark, the colors of the scales were modified for coherence with the adjacent plot.

2. **Line 212: Typo - should be "hopanes" instead of "hopaes"**

   We thank the referee for spotting this typing error. It is now corrected.

3. **Line 225, Multisite PMF: the dimensions of the final matrix should be mentioned, number of variables used in final run is missing.**
   The dimensions of the final matrix were included in the description of the methodology.
   *"For this purpose, in order to combine both datasets as one (EA-LP) the dates of the La Paz dataset were shifted in time by two years and then appended to El Alto's dataset, thus avoiding repeated dates and composing a single input matrix for PMF that respected the natural seasonal variability of the original datasets. The dimensions of the resulting matrix were 185 rows (samples) x 40 columns (species). The multisite approach stands on the hypothesis that the major sources contributing to $PM_{10}$ in both sites are similar and display similar chemical profiles, which has been verified within the single site solutions."*

4. **Line 240, fuel fingerprint: The units in results of chemical analysis of fuel in SI are apparently wrong, must be corrected. Moreover, the reviewer is surprised by high concentrations of arsenic in gasoline in comparison with lead and zinc. Is that really correct?**

   We thank the referee for spotting the missing magnitude correction. Concentrations are indeed 2 orders of magnitude lower and were corrected in the SI as displayed below:

Table S3. Species analyzed from fuel samples collected at La Paz and El Alto

| | Sample# | Al | Cr | Mn | Fe | Co | Ni | Cu | Zn | As | Ag | Cd | Pb |
|---|---|---|---|---|---|---|---|---|---|---|---|---|---|
| | | mg/l | mg/l | mg/l | mg/l | mg/l | mg/l | mg/l | mg/l | mg/l | mg/l | mg/l | mg/l |
| *gasoline* | 1 | 13.7 | 3.96 | 63.0 | 2.81 | 0.05 | 0.23 | 0 | 0.85 | 4.00 | 0.12 | 0.002 | 0.10 |
| *gasoline* | 2 | 8.23 | 3.84 | 67.3 | 0.00 | 0.03 | 0.10 | 0 | 1.68 | 3.47 | 0.09 | 0.003 | 0.10 |
| *gasoline* | 3 | 8.54 | 3.51 | 61.7 | 0.00 | 0.03 | 0.07 | 0 | 1.71 | 3.07 | 0.01 | 0.004 | 0.10 |
| *diesel* | 4 | 14.9 | 9.82 | 0.66 | 0.00 | 0.02 | 0.13 | 0 | 7.85 | 4.00 | 0.01 | 0.005 | 0.25 |
| *diesel* | 5 | 15.8 | 8.81 | 0.41 | 0.00 | 0.02 | 0.13 | 0 | 6.55 | 3.74 | 0.00 | 0.007 | 0.27 |
| *diesel* | 6 | 35.0 | 10.2 | 0.91 | 11.6 | 0.02 | 0.11 | 0 | 7.49 | 4.36 | 0.01 | 0.011 | 0.28 |

   Nevertheless, the ratios between the species are maintained, and are the values measured from the samples collected on site.

5. **Line 250: Fig. number is missing in this line.**

This constitutes a typing error that is now corrected. *"Fig. 2"* We thank the referee for spotting it.

6. **Line 278 -280, equation 4. The equation contents mistakes. Sulphates are corrected for sea salt sulphates, but no sea salt is added. In addition, crustal element mass calculation is a bit strange, why calcium is mentioned twice and only part of other elements are corrected for oxygen, the same is valid for phosphor.**

We agree with the referee, for which a more detailed description of the terms composing the equation was included in the main text. The form of the equation was modified based on a deeper bibliography research, which now includes a sea salt term and the oxygen correction for all crustal elements

*"The reconstruction of the measured PM$_{10}$ mass resulted from the mass closure of the major components of PM, as described in Favez (2010), Putaud (2004), Seinfeld & Pandis (1998), Chan et al., (1997), Pérez (2008), and Cesari et al., (2016). Thus,*

$$PM(recons) = \{(1.8[OC])\} + \{[EC]\} + \{([SO_4^{2-}] - 0.252[Na^+]) + [NO_3^-] + [NH_4^-]]\} + \{2.54[Na^+]\} + \Big\{1.15 \cdot$$
$$\Big((1.89[Al]) + (2.14 \cdot (2.65[Al])) + 1.67[Ti] + (1.4 \cdot ([Ca] - [Ca^{2+}])) + (1.2 \cdot ([K] - [K^+])) + 1.36[Fe]\Big) +$$
$$(1.5[Ca^{2+}] + 2.5[Mg^{2+}])\Big\} \qquad (4)$$

*Where the first curly bracket accounts for the organic matter, the third one accounts for the sum of the mass of secondary inorganic aerosol particles (non-sea-salt sulfate, nitrate, and ammonium), the fourth accounts for sea salt, and the fifth curly bracket accounts for the mass of the main components of crustal material: Al$_2$O$_3$, SiO$_2$, TiO$_2$, CaO, K$_2$O, FeO and Fe$_2$O$_3$ (multiplied by 1.15 to take into account sodium and magnesium oxides), and the mass of unmeasured carbonates. "*

7. **Line 281 – 284: Water absorbed in aerosol and adsorbed on the filter can be also part of unresolved mass.**

We thank the referee for the correction, also brought up by referee #1, which is now corrected.

*"Average PM10 (recons.)/ PM10 (meas.) ratios of 0.91 in El Alto and 0.82 in La Paz were found. The remaining unidentified mass fraction may be attributed to the loss of volatile organic matter and secondary aerosols post-weighing, during the transport of the filter fractions to be analyzed. The difference can also be associated to the presence of non-measured species (i.e. carbonates) or to the adsorption of water in the aerosol particles or the filter (Pio et al., 2013). Moreover A 10% uncertainty associated with the gravimetry measurements could also have a role in the observed difference".*

8. **Line 319 - Dust factor in El Alto, the authors mentioned that sampling site was surrounded with dusty surface. Although the sampling site was on the roof of the building, it cannot be excluded that sampling site is influenced by the local dust. It should be mentioned here.**

We included a sentence to remark what was suggested by the referee.

*"The dust factor has outstanding contributions of 32% in the city of El Alto, becoming the dominant source in this city. Although the volume sampler was placed on the roof of the observatory building, it cannot be excluded that the samples were influenced by the local dust. For La Paz, the vehicular emissions take the lead in terms of percentage contributions (35%). The factors associated to secondary aerosols (secondary sulfate, secondary nitrate, MSA-rich) were responsible for nearly*

*22% and 24% of total PM (La Paz and El Alto respectively), only a slight difference can be observed between the cities except for the nitrate rich profile."*

9. **Line 320 – The factors associated to secondary aerosol … It should be mentioned which factors. Probably sulphates, nitrates and MSA rich factors, but it should be specified.**
   A specification was included in the text.

   *"The factors associated with secondary aerosol particles (secondary sulfate, secondary nitrate, MSA-rich) were responsible for nearly 22% and 24% of total PM (La Paz and El Alto respectively), only a slight difference can be observed between the cities except for the nitrate rich profile."*

10. **"Line 441 – metals such as Cu, Sn, Sn and Pb. Not sure if one of Sn should be Zn or Sb (Zn seems more probable)**
    We thank the reviewer for spotting the typing error. The duplicated Sn was replaced by Sb.

11. **Line 473 – citation F. Amato et al., 2011 should be Amato et al. 2011**

    The format of the reference was corrected

12. **Line 598 – "natural anthropogenic sources" should be "natural and anthropogenic sources"**
    We thank the referee for spotting this typing error that is now corrected: "there is a significant contribution of regional natural *and* anthropogenic sources of PM (Primary and secondary biogenic emissions, and biomass burning)"